# Microsensor Electrodes for 3D Inline Process Monitoring in Multiphase Microreactors

**DOI:** 10.3390/s20174876

**Published:** 2020-08-28

**Authors:** Sebastian Urban, Vinayaganataraj Tamilselvi Sundaram, Jochen Kieninger, Gerald A. Urban, Andreas Weltin

**Affiliations:** Laboratory for Sensors, IMTEK Department of Microsystems Engineering, University of Freiburg, 79106 Freiburg, Germany; sebastian.urban@imtek.uni-freiburg.de (S.U.); vinayaganataraj.sundaram@venus.uni-freiburg.de (V.T.S.); kieninger@imtek.uni-freiburg.de (J.K.); urban@imtek.de (G.A.U.)

**Keywords:** process monitoring, electrochemical sensors, oxygen, hydrogen peroxide, chronoamperometry, microreactor

## Abstract

We present an electrochemical microsensor for the monitoring of hydrogen peroxide direct synthesis in a membrane microreactor environment by measuring the hydrogen peroxide and oxygen concentrations. In prior work, for the first time, we performed in situ measurements with electrochemical microsensors in a microreactor setup. However, the sensors used were only able to measure at the bottom of the microchannel. Therefore, only a limited assessment of the gas distribution and concentration change over the reaction channel dimensions was possible because the dissolved gases entered the reactor through a membrane at the top of the channel. In this work, we developed a new fabrication process to allow the sensor wires, with electrodes at the tip, to protrude from the sensor housing into the reactor channel. This enables measurements not only at the channel bottom, but also along the vertical axis within the channel, between the channel wall and membrane. The new sensor design was integrated into a multiphase microreactor and calibrated for oxygen and hydrogen peroxide measurements. The importance of measurements in three dimensions was demonstrated by the detection of strongly increased gas concentrations towards the membrane, in contrast to measurements at the channel bottom. These findings allow a better understanding of the analyte distribution and diffusion processes in the microreactor channel as the basis for process control of the synthesis reaction.

## 1. Introduction

Hydrogen peroxide (H_2_O_2_) is a clean and very versatile reactive chemical typically used as an oxidizing agent. Because of its excellent chemical properties and sustainable nature, it is widely used in textile, metal, pulp, bleaching, chemical synthesis, and electronics [1,2,3,4,5]. Its green properties originate from the clean decomposition of forming only water, making it one of the key chemicals in a sustainable chemical industry [1]. Although H_2_O_2_ itself is an eco-friendly chemical, the way it is produced is not an ecologically sustainable process from the green chemistry viewpoint. Currently, the major production of H_2_O_2_ is done through the so-called anthraquinone auto-oxidation (AO) process, which alone is responsible for around 95% of the overall H_2_O_2_ production [6,7,8]. The process is complex, requires a lot of energy, and produces toxic, organic waste [3,5]. Considering these disadvantages, the heterogeneously catalyzed direct synthesis of H_2_O_2_ from molecular hydrogen (H_2_) and oxygen (O_2_) provides an attractive alternative process route [9,10,11].

Using synthesis membrane microreactors is one of the most promising direct synthesis methods to produce H_2_O_2_ [10]. H_2_ and O_2_ are flushed separately into the microreactor, in which they diffuse through a gas permeable membrane into a catalytic suspension fluid containing water, ethanol, methanol, or mixtures with catalytic promoters like sodium bromide, or acids like sulfuric acid or phosphoric acid [12,13,14]. There the gases react to H_2_O_2_, leaving only water as a potential byproduct. In principle, using microreactors offers the advantage of enhanced heat and mass transport properties, and therefore increases H_2_O_2_ productivity due to their internal channel dimensions in the sub-millimeter range [15]. Reactor concepts like these can be improved further by combining them with other process intensification techniques like membrane technology among other things [16]. In order to take advantage of the potential benefits of such state-of-the-art reactor concepts, the local concentrations of the reactants, in this case H_2_, O_2_ and H_2_O_2_, needs to be determined to guarantee an optimal control of the synthesis reaction.

Commercially available analytic systems currently only allow for limited assessment of the local concentrations of the reactants H_2_, O_2_ and H_2_O_2_, as they are only able to measure at the in- and outlets of the microreactor. This makes defining and optimizing of important process parameters a challenging task.

Electrochemical sensors, in principle, are a perfect fit for such applications, since they allow for continuous acquisition of analyte concentrations dissolved in the liquid phase, while providing high spatial and temporal resolution and also allowing the precise detection of even small concentration changes, thanks to their high selectivity and sensitivity [17,18,19,20,21,22]. Despite these and other obvious advantages, electrochemical sensors are rarely used in the field of micro process technology. Therefore, in prior works, we developed and used microsensors based on platinum electrochemistry for the amperometric in situ online monitoring of the synthesis educts and products inside the microreactor [21,23]. As shown in Figure 1A, the previous sensor design was integrated at the bottom of the microchannel. The synthesis educt gases are dosed through a polydimethylsiloxane (PDMS)-based membrane which is situated 500 μM above the sensor. Therefore, the gases have to diffuse this distance through the liquid phase before reaching the sensor, while already undergoing the synthesis reaction. This allows only for a limited assessment of the gas distribution and concentration change over the reaction channel dimensions.

In this work, we present an advanced concept of our microsensor plug design, which allows the measurement of the analytes H_2_, O_2_, and H_2_O_2_ inside the microchannel using a novel, protruding electrode wire approach, as shown in Figure 1B. Measurements were performed comparing the flat with the new, protruding wire design in the microreactors. The concept allows not only the measurement along channel length and width, but also in three dimensions (3D) along the channel height. The results showed a significant increase in detected educt gas concentration, depending on the residence time of the liquid phase when measuring closer to the gas permeable membrane.

## 2. Methods and Experimental

### 2.1. Sensor Device Fabrication

The presented microsensor design is based on the electrochemical detection of the analytes using an integrated three-electrode setup. The working (WE) and counter electrodes (CE) consist of platinum wires with a diameter of 300 μM, as it is able to oxidize (H2O2 and H2) and reduce (O2 and H2O2) all analytes which are involved in the synthesis, without requiring any surface modifications by applying chronoamperometric protocols [22,23,24]. Since the liquid phase contains a fixed sodium bromide concentration (4 mM), an on-plug silver/silver bromide (Ag/AgBr) electrode with a diameter of 500 μM is used as pseudo-reference electrode. The deposition of the Ag/AgBr electrode was carried out by anodic bromination of the silver wire as described in more detail in our prior work [21].

To withstand the rather harsh environmental conditions prevalent inside the synthesis microreactor (acidic pH, up to 100 bar of pressure), a robust packaging of the sensors is required. For this purpose, the electrodes were first cut into length, insulated on the outside with heat shrinking tubes (polyolefin, TE connectivity), and soldered into a 1.27 mm pitch connector (Figure 2) The as prepared electrodes were fixed into a Polytetrafluoroethylene (PTFE)-based casting mold, which was then filled with a two-component epoxy (Loctite Stycast 2057, Henkel, Düsseldorf, Germany), leaving a part of the wires unexposed to the epoxy. After complete hardening of the epoxy, the sensor plugs were removed from the mold. In this state, the protruding wires on the sensor plug have an unspecified length. Therefore, in the next step, the sensors were fixed into a mold, which leaves an opening around the free-standing wire part of the plug. This opening is filled with a fast-curing epoxy (Technovit 5071, Kulzer, Hanau, Germany), soluble in organic solvent, which was used as a sacrificial mechanical support for the wires during machining and polishing. The wires were then cut into the exact length needed using a CNC milling machine and subsequently polished to obtain a high quality electrode surface. Figure 2 shows a schematic overview of the fabrication process of the sensor plugs. A pHEMA-based hydrogel membrane, as reported previously [21], was applied to the electrodes for an increase in diffusion limitation and thus more linear range, as well as for the minimization of the dependency of the sensor signal on the flow rate.

### 2.2. Electrochemical Measurements

The electrochemical detection of the reactants was performed utilizing amperometric and chronoamperometric measurement methods by using a three electrode setup, consisting of on-plug WE/CE and Ag/AgBr reference electrodes. The measurements were performed in an aqueous electrolyte with a pH of 3.5 consisting of 0.15 mM H_2_SO_4_ and 4 mM NaBr dissolved in deionized water. By utilizing a chronoamperometric protocols, this system is able to detect H_2_O_2_ through direct oxidation on the platinum electrode surface, while also being able to measure the dissolved O_2_ concentration in the reaction channel by switching to a different potential allowing its reduction on the same electrode. H_2_O_2_ was measured anodically at a potential of 850 mV vs. Ag/AgBr, whereas O_2_ was measured cathodically at −300 mV vs. Ag/AgBr, if not stated otherwise. In all measurements, the chronoamperometric protocol applied both potentials in alternation for 10 s each. All measurements were performed at normal pressure and room temperature. Different concentrations of dissolved O_2_ in the liquid phase of the reactor were adjusted by flushing the gas phase of the reactor with gas mixtures from pressurized air and nitrogen using an IL-GMix41 gas mixing station (HiTec Zang, Herzogenrath, Germany).

## 3. Results and Discussion

### 3.1. Sensor Calibration

Three sensor plug types with different length of protruding wires were installed into the microreactor. After installation, the plugs were calibrated first for different O_2_ concentrations. An electrolyte reservoir was flushed with increasing concentrations of O_2_, regulated by the ratio of a N_2_/O_2_ gas mixture from zero percent O_2_ to pure O_2_. The electrolyte (deionized water with 0.15 mM H_2_SO_4_ and 4 mM NaBr) was constantly pumped through the microreactor with an Abimed Minipuls 3 (Gilson, Middleton, Wisconsin USA) peristaltic pump. To ensure that no change in dissolved gas concentration occurred during the pumping through the microreactor, the corresponding gas concentration was also dosed into the gas inlets of the microreactor. The measurement for each O_2_ concentration was repeated three times. Before each measurement, the electrode was cleaned by a chronoamperometric protocol. Figure 3A shows the resulting calibration curve for the averaged values of the calibration measurements for each of the used plugs. All three plugs show an excellent and repeatable linearity up to O_2_ saturation at 1 bar O_2_ partial pressure (R^2^ = 0.998 and 0.999) with very good sensitivities between −2.088 and −2.859 mA cm^−2^ mM^−1^.

H_2_O_2_ calibration was performed on all three sensors by preparing electrolyte solutions with H_2_O_2_ concentrations of 0 mM, 0.5 mM, and 1 mM. Each electrolyte was flushed for one hour with nitrogen to eliminate any oxygen background in the measurement signal before pumping it through the microreactor. During the measurement, nitrogen gas was also applied through the gas inlet of the microreactor to ensure the complete absence of dissolved O_2_ in the electrolyte. For each concentration of H_2_O_2_, three measurements were performed.

The resulting calibration curve for the arithmetic mean of the three measurements is shown in Figure 3B. All three plug types showed excellent linearity up to 1 mM of H_2_O_2_ concentration (R^2^ = 0.994 to 0.999), very good sensitivity between 0.836 and 1.079 mA cm^−2^ mM^−1^, and a defined zero-point.

### 3.2. Oxygen Measurement at Different Reactor Positions and Sensor Heights

For dissolved gas concentration experiments, the sensor plugs were integrated into the different reactor positions from 1 (close to the inlet) to 8 (close to the outlet). Four plugs were inserted across all eight reactor positions for measuring the dissolved O_2_ gas concentration. One measurement was performed while measuring four plugs simultaneously and the positions were changed afterwards. Each reactor position measurement was performed with two different flow rates of 0.5 mL/min and 2.2 mL/min, resulting in different residence times of the electrolytes inside the microreactor, and depending on shorter or longer times, the electrolyte was exposed to higher or lower amounts of the gas diffusing through the membrane at the top of the channel. The electrolyte reservoir was flushed with air for one hour before and during the pumping through the microreactor. During the measurement, nitrogen was applied to the gas phase of the microreactor. Therefore, the measured O_2_ gas concentration should be the highest at the inlet (position 1) and should decrease over the length of the reactor channel to the lowest value close to the outlet (position 8). Figure 4A shows the used microreactor setup used for the following experiment and gives an indication about the eight sensor plug positions referenced in the following results.

Figure 4B,C show the resulting measured decrease of the O_2_ concentration over the length of the channel and therefore increased exposure time to the nitrogen gas. At a flow rate of 2.2 mL/min, only a small decrease in dissolved O_2_ concentration was visible (from 290 μM to 260–250 μM) as the residence time was too low for a considerable amount of nitrogen gas diffusing into the liquid phase. No noteworthy difference between the different sensor heights could be observed.

When lowering the flowrate to 0.5 mL/min, a clear change in O_2_ concentration over the length of the channel could be observed, as shown in Figure 4B. The increased residence time led to more nitrogen diffusing into the liquid phase, replacing the dissolved O_2_. A distinct difference between the different protruding electrode heights could be observed. Electrodes with wire heights of 265 µm and 235 µm measured the largest change, followed by the plug with wire height of 145 µm. The sensor that only measured at the bottom channel wall measured the smallest change. Comparing how the length of the protruding wires affects the detection of changes in the O_2_ concentration for different flow rates, and therefore different residence times of the liquid phase, it becomes clear how important it is to consider the electrode position in such a multiphase application.

Figure 5A,B show the measured O_2_ concentrations inside the microreactor for different flow rates of the liquid phase. While a decrease of O_2_ could be observed for both sensor designs, distinct differences were visible, especially for lower flow rates (higher residence times), even though the conditions were the same for each measurement. For the lowest flow rate of 0.5 mL/min, the sensor without the protruding wires still measured an O_2_ concentration of 223 µM at position 8 close to the outlet of the reactor. When measuring with a sensor using protruding wires of 265 µm for the same flow rate, at the same position an O_2_ concentration of 80 µM could be observed. This amounted to a decrease of measured O_2_ of 64%, clearly showing that the diffusion of the gas through the liquid phase was slow enough to create a significant concentration gradient over the channel height. By measuring only at the bottom of the channel, one can therefore not directly deduce the reaction conditions at different heights in the channel of the liquid phase inside the microreactor during the synthesis process.

This showed the importance of measuring not only at one position in the microchannel during the process monitoring of the synthesis, but also of assessing the diffusion behavior of the dosed gases by measuring closer to the gas permeable membrane at the top of the reactor channel to be able to more precisely determine the gas distribution and concentration change over the reaction channel dimensions. A future plug design could combine protruding sensor lengths of different height on one plug to be able to always asses the conditions at different distances to the phase-separating membrane to more precisely model and control the synthesis reaction.

Our fabrication process and sensor integration of protruding microelectrodes was not limited to the shown membrane microreactor application. Any situation where concentration gradients in liquids occur, either by parabolic flow profiles in (micro-)fluidics, dosage/removal of dissolved gases through membranes, or consumption/generation of analytes in stagnant media, may be potential use cases. These range from flow chemistry and process engineering over chemo- and biosensors, as well as energy applications, to bioreactors and cell/tissue culture environments.

## 4. Conclusions

We designed and fabricated an electrochemical microsensor device using a novel, free-standing wire approach, which allowed the multiparametric, three-dimensional measurement of the dissolved reactants H_2_O_2_ and O_2_ for in situ process control of the H_2_O_2_ direct synthesis in membrane microreactors. A fabrication process and the robust integration of free-standing microelectrodes into stainless steel microreactors was introduced.

The new concept allowed measurements in the center of a microchannel and along the vertical axis. It was utilized to measure closer to the gas-permeable membrane dosing the process gases into the liquid phase, instead of only being able to measure at the bottom of the microchannel. As shown in our results, a measurement at the channel bottom provides only limited information on the concentration of the gas diffusing into the liquid phase due to the rather slow diffusion of the dissolved gases. Our new approach enables a better understanding of the diffusion and potential reaction behavior of the reactants during the synthesis process.

## Figures and Tables

**Figure 1 sensors-20-04876-f001:**
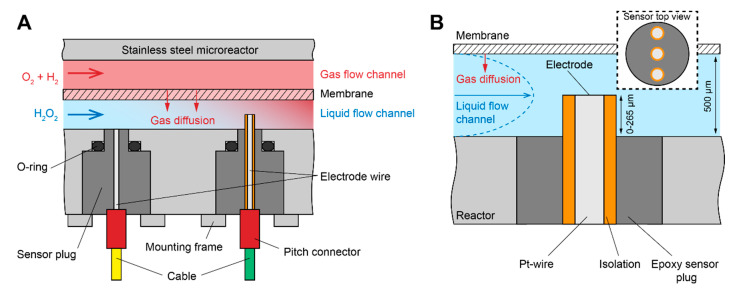
(**A**) Schematic of the membrane reactor principle and sensor placement with the previous, flat sensor design (left) and the new, advanced concept, which allows for measurements in three dimensions (right) inside the microreactor. (**B**) Detailed view of the advanced sensor plug design using a novel, insulated, free-standing wire approach to allow measurements of the synthesis reactants along the vertical axis in the microchannel. An insert schematic of the top view of the sensor shows the multi-electrode setup, which also allows for the measurement of the analyte concentration along the channel width.

**Figure 2 sensors-20-04876-f002:**
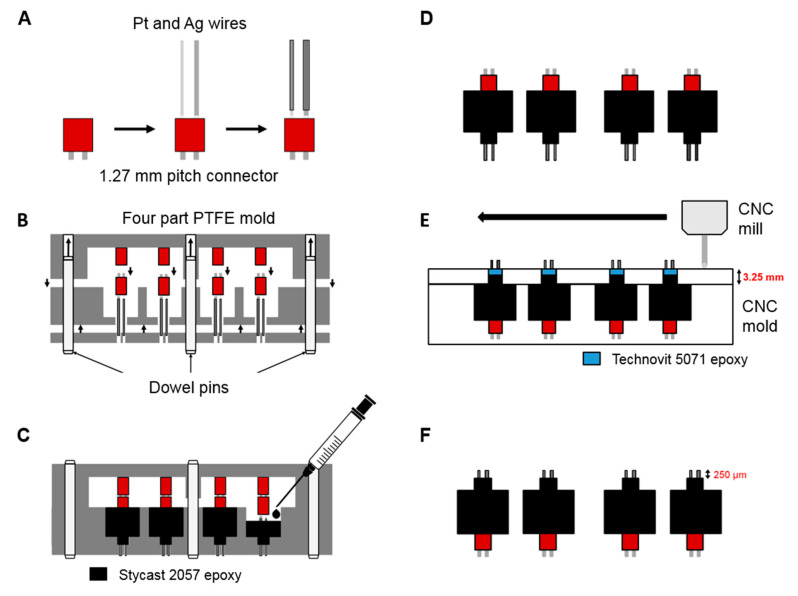
Schematic showing the 3D sensor plug fabrication. (**A**) Pt and Ag wires are cut to a specific length, soldered into a pitch connector, and insulated with heat shrinking tubes. (**B**) The soldered wires are placed into a PTFE mold. (**C**) The mold is filled with a two-component epoxy and cured. (**D**) The sensor plugs with an undefined length of the free-standing electrodes are removed from the mold. (**E**) The free-standing wires are cut into the required length by CNC milling, while being supported by a sacrificial epoxy layer. (**F**) The sensor plugs are polished and the sacrificial epoxy layer is removed by dissolving it with acetone, leading to free-standing, insulated wires with electrodes exposed at the tip.

**Figure 3 sensors-20-04876-f003:**
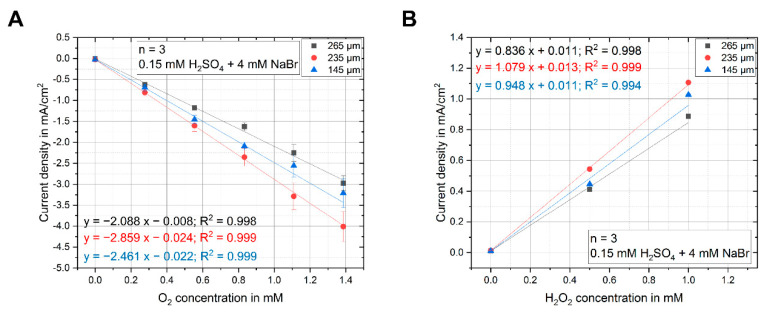
(**A**) Calibration curves in the membrane reactor for the O_2_ reduction at −300 mV vs. Ag/AgBr showed excellent linearity up to saturation at normal pressure (R^2^ = 0.998–0.999). (**B**) Calibration curves for the detection of H_2_O_2_ for concentrations up to 1 mM showed excellent linearity (R^2^ = 0.994–0.999) and high sensitivity. The error bars represent the standard deviation from different measurement runs.

**Figure 4 sensors-20-04876-f004:**
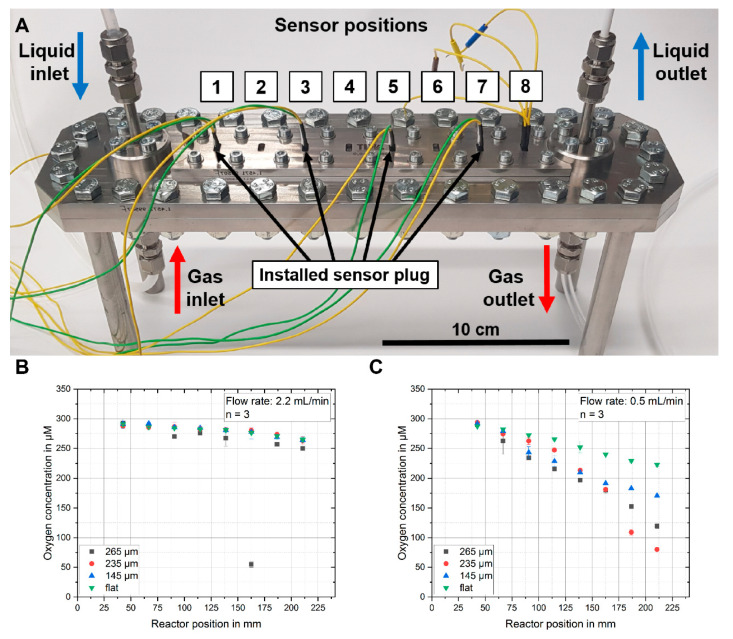
(**A**) Multiphase membrane microreactor with the measurement setup, showing the numbering and the corresponding positions of all electrode plugs (from 1 close to the inlet to 8 being close to the outlet of the microreactor). (**B**) Oxygen concentration measured over the length of the microreactor for a flow rate of the liquid phase of 2.2 mL/min (from position 1 close to the inlet to position 8 close to the outlet). The electrolyte was air-saturated. (**C**) The same measurement performed at a lower flow rate of 0.5 mL/min resulted in an accordingly longer exposure time of the liquid phase to the nitrogen gas diffusing through the membrane. The error bars represent the standard deviation from three measurements in the same run.

**Figure 5 sensors-20-04876-f005:**
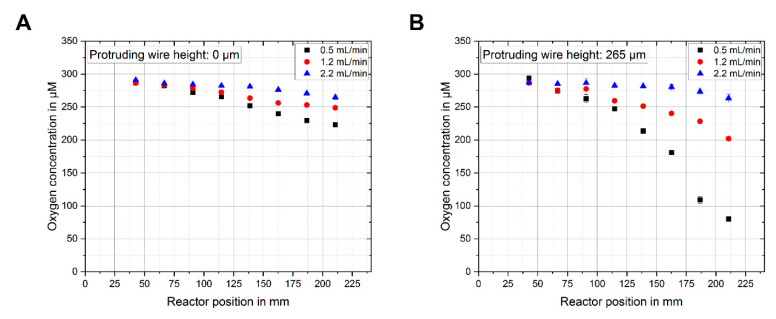
(**A**) O_2_ concentration measured with a sensor without the protruding wire design at eight different positions inside the microreactor for the flow rates of 0.5, 1.2, and 2.2 mL/min. The increased residence time leads to a higher change in O_2_ concentration due to the dosage of nitrogen through the membrane. (**B**) The same measurement performed with the new sensor design with a protruding wire length of 265 µM, showing a stronger removal of the dissolved O_2_ closer to the membrane. The error bars represent the standard deviation from three measurements in the same run.

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
