# Peer review of "Microsensor Electrodes for 3D Inline Process Monitoring in Multiphase Microreactors"

_sensors, 2020, doi:10.3390/s20174876_

Round 1

Reviewer 1 Report

Manuscript describes a development an electrochemical microsensor for the monitoring oxygen and hydrogen peroxide concentration in a membrane microreactor. The developed microsensor is upgrade of the previously developed sensor. Reported microsensor has ability to perform detection at various positions (height) in microreactor. The article is written and prepared as technical report. The article is written very concise and briefly in engineering style and consist all relevant information and description of methods and results for such kind of an article.

The suggestions and the comments to the authors are given as follows.

GENERAL COMMENTS:

  • using term "3D" in the context of developed sensor is not clear. The sensor has ability to change position in one direction. Although, it is wire (2D) that can be moved by height (1D), it seems that using 3D is a bit
  • in Abstract authors have a statement that article present microsensor for monitoring “hydrogen peroxide, oxygen and  hydrogen  concentrations” (lines 12-13). However, in article hydrogen concentration monitoring is not reported.

SPECIFIC COMMENTS:

  • line 13: “in situ” must be written in italic.
  • line 89: reference or decription for preparation of Ag/AgBr should be given.
  • Figure 2: It is not clear that prepared microsenor contains three electrodes.
  • line 132: “Table 1” is missing in Manuscript.

My apologies to poor English.

Best Regards

Reviewer 2 Report

The authors hoped to develop novel 3D microsensor electrodes for monitoring the hydrogen peroxide, oxygen and hydrogen concentrations at any time. However, the idea is not so correct and the  manuscript should be rejected by the journal:
1. The full name of "3D", that is , three dimentional, should be provided in the Abstract;
2. The references should be formatted accordingly;
3. H2O2 and hydrogen peroxide are randomly presented in the manuscript, why?
4. For "...transport properties and therefore increased hydrogen peroxide productivity, due to their internal...", "increased" should be "increases";
5. For "Electrochemical sensors, in principle, are a perfect fit for such applications...", some related references should be presented;
6. For the concept of "The concept allows not only the measurement along channel length and width, but also in 3D along the channel height" in Fig. 1B, the schematic illustration is not meaningful;
6. Platinum wire used as a working electrode is too obvious, the 3D concept is really doubtable and not new;
7. "chronoamperoemtric protocol" should be "chronoamperometric protocol";
8. "regulated by the ratio of a N2/O2 gas mixture from zero percent O2 to pure oxygen"? The lower the gas concentration is, the higher the uncertainty is!
9. The measurement at the bottom of the microchannel is feasible, while the measurement in the center of a microchannel and along the vertical axis couldn't be performed correctly because the concentrations of the two-phases are not stable in different heights.
10. How to confirm the results obtained by the 3D microsensor electrodes are correct?

Reviewer 3 Report

This manuscript describes a micro-electrode system for inline monitoring in a multiphase microreactor.  Specifically, this microsensor system measures hydrogen peroxide (H2O2) generated by oxygen and hydrogen direct reaction inside a membrane microreactor.

This manuscript is a modified designed of the electrochemical sensor system based on previous research accomplished by the authors (cited references #18 and # 19 as well as the design shown in Figure 1A) The modification of the sensor system described in this manuscript is shown in Figure 1B. Basically, this modification employs a protruding electrode allowing the measurement of the gases not only  in the bottom of the microchannel of the microreactor but also along the vertical axis of the microchannel.  

Figure 3 of this manuscript shows the calibration curve of the oxygen reduction and the H2O2 concentration measured by the modified electrode system. However, in the actual experimental results as shown in Figures 4 and 5, only the oxygen concentration as a function of the position and the flow rate are given.  It is necessary to provide experimental measurement of the H2O2 concentration generated at these sensor placing position and the flow rate, since H2O2 is the main product of this synthesis process.  The effect of the oxygen concentration changes due to the positions of the sensor in the microchannel on the production of H2O2 in this synthesis process is not known nor verified.  

Ag/AgBr was used as the reference electrode in this three electrode system.  In the absence of any Br ions in the test medium, Br ions from the Ag/AgBr reference electrode will diffuse out from a higher concentration of the surface of the reference electrode to the zero concentration of Br ions in the test medium. Thus, the stability and the length of useable time for the reference electrode of this sensor system needs to be assessed and discussed.  

The pH value will change as the H2O2 produced in this synthesis process.  It will be meaningful for the authors to assess and discuss if any significant change of  the pH value in the test medium at different stages of the oxygen reduction occurs.  

Round 2

Reviewer 2 Report

It can be accepted

Reviewer 3 Report

The revised manuscript addresses the concerns and limitation of the research in the original submission.  The authors have addressed the issues of concern satisfactory.  The manuscript is acceptable for publication in its revised version.